# A Model for the Spread of Infectious Diseases in a Region

**DOI:** 10.3390/ijerph17093119

**Published:** 2020-04-30

**Authors:** Elizabeth Hunter, Brian Mac Namee, John D. Kelleher

**Affiliations:** 1School of Computer Science, Technological University Dublin, D24 FKT9 Dublin, Ireland; 2School of Computer Science, University College Dublin, D04 V1W8 Dublin, Ireland; brian.macnamee@ucd.ie; 3ADAPT Research Centre, Technological University Dublin, D24 FKT9 Dublin, Ireland; john.d.kelleher@tudublin.ie

**Keywords:** infectious disease, epidemiology, agent-based model, centrality, simulation

## Abstract

In understanding the dynamics of the spread of an infectious disease, it is important to understand how a town’s place in a network of towns within a region will impact how the disease spreads to that town and from that town. In this article, we take a model for the spread of an infectious disease in a single town and scale it up to simulate a region containing multiple towns. The model is validated by looking at how adding additional towns and commuters influences the outbreak in a single town. We then look at how the centrality of a town within a network influences the outbreak. Our main finding is that the commuters coming into a town have a greater effect on whether an outbreak will spread to a town than the commuters going out. The findings on centrality of a town and how it influences an outbreak could potentially be used to help influence future policy and intervention strategies such as school closure policies.

## 1. Introduction

Public health, and in particular the risk of infectious diseases, is major global security risk. With individuals travelling globally, infectious diseases can spread quickly worldwide. The SARS outbreak in 2003, the H1N1 pandemic in 2009, the MERS outbreak starting in 2012 and the COVID-19 pandemic in 2020 all bring this into focus. As a pandemic can threaten not only lives but our daily routines, it is important to look at at how we can better prepare for the next outbreak.

At a more local level, global travel and lower-than-desired vaccination rates mean that outbreaks of diseases are still common in countries where those diseases are no longer endemic. For example, measles has been declared no longer endemic in Ireland, but in the first 3 months of 2018 there were 59 measles cases reported in Ireland from three different outbreaks [1] (the World Health Organization (WHO) defines a measles outbreak as two or more linked cases of measles [2]). Within Europe, the first three months of 2018 saw 18,325 measles cases in 36 different countries with 23 deaths secondary to measles [1]. With a more global population, infectious diseases are more of a threat than ever. To help design prevention and response strategies for future outbreaks it is important to understand how diseases spread and to incorporate this understanding into models as accurately as possible.

There are many aspects of infectious disease spread that can help inform the best policies to prevent an outbreak from occurring or to stop an outbreak before it it spreads. One such factor is the susceptibility of the population of an urban environment (such as a town) to an outbreak. Hunter et al. [3] use openly available data to create an agent-based model capable of simulating the evolution of outbreaks within a population of a town when the town is considered in isolation. This simulation models several different factors and their interactions that influence the likelihood that a measles outbreak would take hold in a town; such as population size, density, and the percent of susceptible individuals within the population. Although information on isolated towns does allow us to analyze the importance of a large number of variables; the fact that the model only considers towns in isolation means that it omits important factors such as travel and commuting patterns between the populations of different nearby towns.

A larger model that considers a network of towns in a given region would be better suited to understanding the dynamics of an outbreak and the true susceptibility of a given town in the network to that outbreak. This better understanding would also help in designing more appropriate response and prevention strategies. In this article we scale up the Hunter et al. [3] model to the regional level focusing on counties in Ireland and look into the level of detail necessary to increase the size of the population being modelled. A larger scale model will allow us to learn more about how an outbreak will spread between towns which could be an important factor in stopping an outbreak before it spreads. A larger scale model can also help to identify towns that might have a higher susceptibility to an outbreak than others. One factor that we think might play an important role in both the spread of an outbreak and the susceptibility of a given town is the centrality of the town in the network. In this paper, we start by giving a brief overview of epidemiology models focusing on agent-based models then discuss scaling up the Hunter et al. [3] agent-based model, and finally run an experiment to determine the importance of the centrality of a town on its susceptibility to an outbreak.

### Epidemiological Models

To better prepare for an outbreak of an infectious disease epidemiologists and public health officials can use modelling. Although there are many kinds of epidemiological models in the literature, in this paper we focus on infectious disease models. Infectious disease models help to understand the spread of an infectious disease through a population. There are multiple ways to model this spread, the two main methods are equation-based and agent-based models. Both methods have advantages and disadvantages but one of the most important advantages of agent-based models is their ability to capture emerging results and interactions between agents’ actions and other characteristics of the population that help to drive the course of an outbreak that equation-based models do not [4]. For that reason this study focuses on agent-based models.

#### Agent-Based Models for Infectious Disease Epidemiology

Agent-based models are a type of computer simulation that are made up of agents and an environment. Agents can interact with each other and their environment and their actions are controlled by a set of coded rules. These rules allow agents to make decisions that determine their actions: the actions can be as simple as defining which direction an agent will move in based on some simulated perception or, the actions can be more complicated such as searching for agents with certain characteristics within a given radius and socially interacting with them [5]. Because agents are able to make their own decisions that determine the course of the model, agent-based models can capture unexpected aggregate phenomena that result from combined individual behaviours [6]. These aggregate phenomena combined with the ability to create heterogeneous agents, social networks, and mixing patterns and the ability to capture accurate disease dynamics mean that agent-based models are becoming increasingly popular as a tool to model the spread of infectious diseases [7]. To realistically model an outbreak, and to be useful in real world scenario, an agent-based model needs to model characteristics of a disease (such as infection rates), as well as agents’ characteristics, their environment and their transportation [8].

The use of real world data allows one to create a model that represents a real population or a real environment. Models that are created using data, such as Rakowski et al.’s [9] model of Poland and Crooks and Hailegiorgis’s [10] model of a refugee camp, can be directly applied to a real population and can help to create and shape policies concerning infectious disease. It is, however, possible to capture the dynamics of a system, such as the spread of an infectious disease, without the use of data. For example, the Dunham model [11] does not use any data to set up their population or run the model. This, however, leads to a disadvantage concerning applicability. A model that does not use any data can be used to study general disease dynamics but does not include characteristics of a given population that might make it more or less susceptible to an infectious disease outbreak and so cannot provide information on how a specific city, county, or country might respond to an outbreak.

## 2. Methods

The Hunter et al. [3] model uses openly available data to model towns in Ireland. The model is an agent-based model that was designed to simulate the spread of measles through an Irish town; however, the towns are considered in isolation with no commuting between towns. The model was validated in several ways: comparing the basic disease dynamics to expected disease dynamics, doing a sensitivity analysis on several key parameters and comparing the results to real data from measles outbreaks in Ireland. In this work, we focus on scaling up this model, from modelling towns in isolation, to take into account the interactions among populations in different towns within a region. In the following methodology sections we break our model down into the four main components of an agent-based model outlined in Hunter et al. [8] and discuss the data used to create each component along with the assumptions necessary to scale up the model and the validation of these assumptions. These assumptions typically involve reducing the fidelity of different components of the model. We then discuss an experiment that is done using the scaled up region model to look at how the centrality of a town influences the spread of an infectious disease through a network. The results of the experiment will be presented in Section 3.

### 2.1. Scaling Up the Town Model to a Region Model

We take the following methodological approach in scaling up the model. For each of the four main components of an agent-based for infectious diseases (environment, society, transportation and disease) we consider assumptions that should be made in scaling up the model most of which involve reducing the fidelity of the model. Each assumption needs to be appropriately validated so that we know the changes and assumptions made did not change the underlying dynamics and function of the model. As the Hunter et al. [3] model has already been validated we aim to show through running simulations that the results of the scaled up reduced fidelity model are not drastically different from the model we already validated. The following sections discuss the assumptions made to scale each of the four main components of an agent-based model for infectious diseases and the validation of those assumptions. Once all of the assumptions are made we additionally validate the entire regional model comparing the results of the town model to that of the regional model focusing on two specific towns.

#### 2.1.1. Environment

Agent-based models allow for a high level of detail. Each agent can have as many individual characteristics as desired and similarly the environment can be rich with detail. However, the greater detail the more computational power needed to run the model and the longer it will take to run. Our aim in scaling up the model is to reduce the fidelity of the environment without greatly influencing the results of the model. The idea is that there is a level of detail in the model that influences our results and there is a level of detail beyond which the results are not affected. In the Hunter et al. [3] model, the environment is made up of small areas and agents can move through the small areas and the town. Small areas are the smallest geographic area that the Irish census is aggregated over. Each small area contains between 50 to 200 dwellings [12]. Residential, commercial and community spaces are designated using zoning data and are used to determine the location of agents’ houses, workplaces and movements within a small area.

In the reduced environment fidelity version of the model, we decrease the environmental space of our model. Instead of having agents live and move through a small area, each small area is represented by a single point in the model. Each environmental point or *patch* in the model that represents a small area has information about that small area that any agents in the small area can access. This information includes the number of primary and secondary schools in the small area, the number of workplaces and the distances between the small area and every other small area in the model.

To test the effect of reducing the environmental fidelity of the model on the model’s output, we run the model for two towns Schull and Tramore 300 times with the original environment and 300 times with the reduced environmental fidelity. All other parameters are held constant and the results are compared. Table 1 shows the percent of the 300 runs that lead to a measles outbreak for both Schull and Tramore and the 95% confidence intervals for both. From the results we can see that the abstraction of small areas to points did not cause the results between the two versions of the towns to vary significantly - the confidence intervals for the detailed environment and the reduced environment overlap. From this result we argue that we can use the reduced environmental version of the model without significantly altering the model.

#### 2.1.2. Society

The society of the model is created using Irish Census data from the CSO [12]. The CSO data is at the small area level. As mentioned in the previous section, small areas are geographic census areas that contain between 50 to 200 dwellings. They are the smallest area over which the Irish census data is aggregated. For each small area we create a population that reflects the population statistics of that small area including age, sex, household size and economic status. Irish vaccination data are used to determine the percentage of each age group that received vaccinations for the infectious disease being modelled. For example, if 90% of 1 year olds in Ireland had been given the MMR vaccination in 2011 and we are running a model for 2012, we give each agent in the model with an age of 2 a 90% chance of having been vaccinated. If an agent is vaccinated they are given a 97% chance of being immune to the disease. This takes into account vaccination failure and is based on the vaccine effectiveness rate for measles [13]. Half of the agents with age less than 1 are given immunity to measles to mimic passive immunity infants receive from their mothers [14]. For any agents that have an age corresponding to a vaccination year not in our data we give a 99% chance of being immune. Prior to vaccination campaigns the majority of the population would have either had or been exposed to childhood diseases, such as measles, leaving them immune in later life. This vaccination method is tailored to measles, to simulate a new or emerging disease we would have to use different assumptions about existing immunity levels.

Social networks are included in the model. These are created based on the simulated society and the schools and workplaces that agents are assigned to in the model. Agents have a family social network that is made up of any agents living in their household. Agents also have a work or school social network that is made up of other agents in their workplace or their school and students are given an additional social network which is a class network that is made up of agents who are in their school and of the same age. Social networks help to determine the contacts an agent has in the model.

In the Hunter et al. [3] model, the population and thus the initial conditions vary slightly from run to run. For each run the model recreates the population again using the same probability distributions. This method allows for variation in the synthetic population and does not settle on a specific version of the population when the exact actual population is unknown. Although it might capture variability in the runs due to one particular set-up being more susceptible than others, there are some disadvantages of running the model this way. The first is time: model set-up can often take a large part of the runtime of the model. Holding a population constant can allow for a speedier set-up as the model does not have to make recreate the model environment on each run. In addition, it increases the variability in the output making it difficult to attribute the difference in the output from the runs to agents’ actions, or societal interventions, versus the variability of the disease itself. An alternative to this is creating the population once and then using the exact same initial population for each run. If we are attempting to show how different interventions such as vaccination rates influence our results, holding the population steady allows us to more accurately attribute changes in our results to the interventions considered. Thus, a constant population between simulation runs would be preferable. However, the effects of holding the population constant on the model results should be investigated in order to determine what impact holding the population steady will have on the model output.

To test the effect of holding our population constant we take two towns in Ireland, Schull and Tramore, both towns used in the Hunter et al. [3] model, and run the model 300 times changing the initial set-up each time and 300 times holding the set-up steady. All other parameters are held constant. The results are then compared between the sets of runs. We look at the percent of runs that lead to an outbreak and the distribution of agents infected across runs. We do not expect the results to be exactly the same when the population is held steady versus when the population changes as we will not capture all possible distributions of the town. We do, however, expect that the results should be similar with the results for the changing population showing more variability when compared to the runs with the population held constant. Table 2 shows the percent of runs leading to an outbreak for each version of Schull and Tramore and the 95% confidence interval for each percent.

From the table we see that our results do not vary significantly when we change the model from creating a new population every run to keeping the same population. The results are broken down further by looking at the summary statistics for the number of agents infected each run which can be seen in Table 3.

From the distributions it is clear that there is a difference in the results. When the population changes with each start there is a larger variation in the results we see this with a higher standard deviation in both the Schull and Tramore models as well as higher 3rd quartiles. This larger variation in results is what is expected as the varying initial conditions are likely capturing some set-ups that are more susceptible to an outbreak than others. However, our results for the percent of runs that lead to an outbreak do not change significantly between the two versions as can be seen from the confidence intervals in Table 2.

#### 2.1.3. Transportation

As we described in Section 2.1.1, the agent-based model described in this paper differs from Hunter et al. [3] by abstracting away from the small areas level of detail, and so agents do not move within a small area (although, importantly for this work, small areas may now be located in different towns) and the only transportation that occurs is between small areas. Each small area is represented by a single environmental unit or patch in the model. Within a small area, all agents in the small area are physically coded in the same patch in the model but the agents keep track of their more abstract location within the small area. For example, an agent will know if they are at home, at work, at school or in the community and can differentiate between being in these different locations even though they will remain on the same patch. Agents move between small areas but do not move around within a small area. However, all agents in the same small area are not in contact with each other at all times. Instead a variety of factors determine if two agents come into contact with each other. First, is the agent’s location, an agent at home will not come into contact with another agent who is at work. Second is an agent’s social networks. An agent will have a higher chance of coming into contact with a member of their family network in the community then a member of their class, school or work network with whom, in turn, they have a greater chance of coming into contact with than an agent who is not in any of their networks.

Movement between small areas is modified for the scaled up version of the model. In the town model, agents move in a straight line between their current location and their destination. When they are deciding on their destination within the community the agents will choose randomly from the possible community spaces in town and move there. Although this is an acceptable assumption for a smaller town, when the size of the area being considered increases the assumption does not hold as well. Moving from one side of a town to another in the space of one hour is not unbelievable; however, moving from one side of a county to another in a short period of time is much less likely. To account for this the agents use a gravity model to choose their next location. Gravity models are a type of transportation model that are based on Newton’s gravitation model. A traditional gravity model gives the interactions between two location pairs and determines those interactions based on the characteristics of a location and the distance between locations [15]. In the model, agents move between home and school or work at certain predetermined times and will return home at predetermined times. On weekends, summers for students and after school or work hours agents movements through the community are determined by the gravity model. The probability of an agent moving to another small area is proportional to the population density of the small area, an area with more agents is more attractive, and inversely proportional to the distance to the small area from the agents current location, areas that are farther away are less attractive. We feel that this transportation model provides a more accurate model of movement within a larger area than that in the original town model.

To capture accurate commuting patterns when agents are not moving around the community the CSO Place of Work, School or College - Census of Anonymity Records (POWSCAR) data is used [16]. This dataset provides information on the commuting patterns of people in Ireland and gives the number of people that commute from one electoral division to another. Electoral divisions are the census geographic area one step above the small areas.

#### 2.1.4. Disease

The disease model is based on a compartmental Susceptible, Exposed, Infected, Recovered (SEIR) model. Where agents start in one of four different compartments, they are either susceptible, exposed, infected or recovered and based on their actions they move between the compartments. The disease model used in this work remains unchanged from the Hunter et al. [3] model. When an infectious agent comes into contact with a susceptible agent they have a given chance of passing the disease to the susceptible agent. If they do pass on the disease, the susceptible agent then moves to the exposed state for a given period of time before moving to the infectious state. An agent will remain in the infectious state for a set period of time before recovering. Once recovered, agents cannot be reinfected.

In the current version of the model, the disease dynamics are set to mimic measles. An individual will stay in the exposed state for an average of 10 days [13]. The time an agent remains exposed is determined for each agent from a normal distribution with a mean of 10 and a standard deviation of 0.5. Once infectious, an individual remains infectious for an average of 8 days [13]. The time an agent remains infectious in the model is determined for each agent from a normal distribution with a mean of 8 and a standard deviation of 0.5. The infection rate, the percentage chance that a susceptible agent will be infected after contact with an infectious agent, is determined using the basic reproductive number $R_{0}$ for measles (12–18) [13]. The basic reproductive number is the number of individuals infected by one infectious individual in a completely susceptible population and is a standard measure of transmission of a disease. The parameter is made up of three components, number of contacts per unit time (*c*), the transmission probability per contact (*p*), and the duration of the infectiousness (*d*) [17]. As we can determine the number of contacts per unit time from our model and we know the duration of the infectiousness, we can determine the transmission probability per contact. Although the model in our experiment is based on measles, changing the parameters described here can allow other infectious diseases to be modelled.

#### 2.1.5. Testing the Scaled Up Model

The extensions and variations of the Hunter et al. [3] model that we described above have been designed to enable us to scale the model up to simulate a geographical region, at a county scale, that contains multiple towns. The key differences between the Hunter et al. [3] model and the scaled up model are outlined in Table 4.

As an initial validation of this scaled up model we decided to compare the results for simulations of outbreaks within two towns in the scaled up version of the model with the results for simulations of outbreaks in the same two towns when the towns were isolated within the simulation. Our expectation was that if the scaled up simulation was working appropriately then the outcomes of the simulations under these different conditions should be somewhat different but not drastically so. If our simulations show drastic differences there may be a problem with how we scaled the model and the assumptions we made would have to be re-checked. This initial validation test is an important step in the methodology of creating an agent-based model. Determining the best method to validate an agent-based model can be difficult because there is no set validation methodology [18]. One of the methods that is used to validate agent-based models is comparing the results of the model to a simpler validated model [8]. Skvortsov et al. [19] validate their agent-based model for infectious diseases in Australia by comparing the disease dynamics of their model to the disease dynamics produced by a SEIR equation-based model. We propose doing something similar and comparing the results of our scaled up region model to the validated town model.

To test and validate our scaled up model we run the region model for the county of Leitrim in Ireland. Based on the 2016 census the county has a population of approximately 32,000 people over an area of 1590 km^2^. The county is made up of 173 small areas. We choose two towns in Leitrim, made up of multiple small areas, and compare the results from the county model to the town model. The idea is that we want the town model to be somewhat stable but still be influenced by being connected to the networks of towns in the county. For each of two separate towns in Leitrim, Manorhamilton and Kinlough, we run the county model 300 times with the outbreak starting in that town. For each run we examine the agents who are sick within the town (i.e., in this experiment we do not consider agents from outside of the town who are sick). We only compare these results to the results for the town model for Manorhamilton and Kinlough (in which the towns are modelled in isolation) and an additional town model that allows for the agents to commute out of the town but does not allow agents to commute into the town (i.e., the only agents in the model are those that live in the town).

The measure we look at for each of our runs is the percentage of runs that lead to an outbreak in the town. We use the WHO definition of a measles outbreak, which is two or more linked cases of measles. For our models we consider an outbreak to be any run where the initial case infects at least one other agent. Table 5 shows the results for the two towns.

From the results we can see that the town only model that allows for commuting results in fewer outbreaks than the town only model where agents cannot leave the town. This makes sense as if the infected agents are commuting outside of the town, once they are outside the town they do not come into contact with other agents and thus cannot spread the disease until they return to the town. In addition, the county model results for both towns are somewhere between the completely closed town model and the town model with commuting. Again, this makes sense as in the county model the agents are not restricted to staying within their town so there is a smaller chance of an outbreak in the town in the county model because in some cases the infected agent will commute out of the town and take the infection with them. The outbreak percentage is, however, higher than for the town model that allows commuting because there are other agents in the model who can become infected keeping the outbreak going. We take this as a sign that the county model is working as it should be. Based on our analysis and validation work in this and previous sections, in the scaled up regional model we will use the model characteristics outlined in Table 4: the environment is created using a single patch for each small area, the society is kept constant each run and transportation is based on a gravity model.

### 2.2. Examining the Role of Town Centrality in the Region-Based Model

After showing that the modelling of towns within the scaled up region-based version of our model is relatively stable we are able to run experiments on the county regional model itself. Studying an outbreak on a network of towns allows us to study how the outbreak propagates through that network and what different factors influence that propagation. In particular, we look at centrality, both the centrality based on the number of agents commuting into and out of a town (degree centrality) and the geographic centrality of the town to all other towns in a network (closeness centrality), and how the centrality of a town influences the spread of an outbreak from a town and the spread of an outbreak into the town. There are several types of centrality and we do not restrict our analysis to just one but instead look at multiple types of centrality and how they interact.

We use the scaled up regional model to do two different experiments to look at how the centrality of a town in a network influences the spread of infectious disease in that town: in the first experiment, we the outbreak starts in a randomly selected small area within the county and then we look at where the outbreak spreads and how many outbreaks occur in each individual town; in the second experiment the outbreak starts in a given town and we again look at where the outbreak spreads. The two experiments are done to determine if the outbreak starting in a given town has an effect on the spread of the infectious disease. Each experiment is run 300 times to account for the stochasticity in the model. We use the county of Leitrim for the model and consider 16 different towns in Leitrim. The towns are: Ballinamore, Carrigallen, Cloone, Dromahair, Dromod, Drumkeeran, Drumshanbo, Drumsna, Fenagh, Keshcarrigan, Kinlough, Leitrim, Lurganboy, Manorhamilton, Mohill and Tullaghan. Characteristic of each town including the population size, area, the number of small areas the town is made up of and the percent of the population that are students can be found in Table 6.

#### 2.2.1. Centrality

When modelling the county or region, we model the network of towns that make up the region. Each town is considered a node in the region and links between two towns are created if at least one agent commutes between the two towns. The strength of the link is determined by the number of agents commuting between those two towns. We also consider the real distance (km^2^) between towns in determining how a town fits into the network.

Degree centrality is the main type of centrality that is used in the experiment. Degree centrality is defined as the number of links between each point in the network, but degree centrality can be defined in multiple ways: total degree centrality includes all links in to and out of a node, in-degree centrality only counts the links going into a node and finally out-degree centrality only calculates the links going out of a node. To get a full picture of how degree centrality effects an outbreak we look at all three versions, which allows us to determine if it is the individuals coming into a town, commuting out or a combination of both that influences the spread of an outbreak.

To account for the number of agents coming into and out of a town a weighted degree centrality is used. The weighted degree centrality is calculated using a product of the number of links and the average weight of the links adjusted by a tuning parameter. Equation (Equation 1) shows the formula for weighted degree centrality with $DC_{i}$ being the centrality of town *i*, $k_{i}$ the number of links into the town, $s_{i}$ the number of agents commuting into or out of the town and α is the tuning parameter.
(1)$$DC_{i}=k_{i}*{\left(\frac{s_{i}}{k_{i}}\right)}^{\alpha }$$

The tuning parameter is used to determine the strength of the weight and the importance of individual link strength: when the tuning parameter is less than one the centrality measure favours more links into the town. If the total number of commuters is fixed a town with more links will have a higher centrality than a town with fewer links. When the tuning parameter is greater than one the centrality measure favours fewer links into the town. If the number of commuters is fixed a town with fewer links will have a higher centrality compared to a town with more links [20]. For the purpose of this study we use an α less than one and set it at 0.5.

If in-degree centrality is calculated instead of out-degree centrality, the weights are only those agents commuting into a town and the links are only number of different towns agents commuting into the town are coming from. Similarly for out-degree centrality weights are only those agent commuting out of the town and links are the paths those agents take.

As degree centrality is not the only factor influencing an outbreak we also look at closeness centrality. Closeness centrality is a measure for how physically central a town is within the network of towns and real world distances are used to calculate the closeness centrality [21]. The formula is:(2)$$CC_{i}=\frac{V-1}{\Sigma (distance\left(v_{i},v_{j}\right))}$$
where *V* is the total number of nodes in the network, and $v_{i}$ and $v_{j}$ represent individual nodes in the network. Here nodes are towns and the distances are calculated from the center of one town to the center of the next.

Table 7 shows the normalized centrality for each of the sixteen towns. The closer to zero the centrality is the less connected a town is in the network. Looking at the the centralities for each town we can see that using these measures, Tullaghan is the least central town. Manorhamilton, has the highest total degree centrality and in-degree centrality but a lower out-degree centrality and closeness centrality. Keshcarrigan is the town with the highest closeness centrality but has middle values for degree centrality. One pattern that can be seen across many of the towns is that a lower in-degree centrality is paired with a higher out-degree centrality and vice versa. Although it is not the case for all towns, this seems logical as it is more likely that people living in a town where there are a large number of commuters coming in will not have to commute out of the town. Similarly, if there are only a small number of commuters coming into the town then it seems more likely that the people living in the town will commute out of town.

#### 2.2.2. Town Similarities

When comparing the results for different towns it is impossible to say which factors of the towns lead to different results. Two towns with different centrality might also have a different population size, town structure etc. and thus the question should arise which factor is actually leading to the difference. In order to attribute most of any difference in results to the difference in centrality we calculate a euclidean distance between each town. This allows us to choose similar towns across several measures so that we are able to better credit any difference in results to centrality. Each town is represented by a vector of quantitative characteristics: population size, town area (km^2^), population density, number of small areas that make up the town, the number of secondary schools, the number of primary schools, the percent of susceptible agents in the town and the percent of agents who are students in the town. All categories except for the number of secondary schools and number of primary schools are standardized. The euclidean distance is then calculated between each of the 16 towns so that we can compare results between similar towns. Figure 1 presents a distance matrix which visualizes which towns are similar based on these categories. The lighter the square the more similar the towns are and the darker the more dissimilar. For example, we can see that towns such as Manorhamilton and Mohill are similar in terms of the profiles measured by the similarity matrix in Figure 1 and are also similar in total and in-degree centrality as seen in Table 7. Dromahair is also similar to Manorhamilton and Mohill based on the similarity matrix; however, has a very different total and in-degree centrality to both Manorhamilton and Mohill but has a similar closeness centrality to Manorhamilton. These similarities and differences help us to select towns so that we can more easily understand the differences in the results of the model.

## 3. Results

The following sections describe the results found from the experiments on the influence of town centrality in a network on the spread of infectious diseases described in the Methodology Section 2.2. For each set of experiment we calculate the percent of runs (The model is run 300 times for each set of initial conditions so that the average results calculated across the runs accounts for variations between individual runs due to stochasticity. The only variation in the initial conditions between runs is the starting location of the outbreak) that lead to an outbreak (two or more cases of measles) occurring in the town. Table 8 shows the percent of runs that lead to an outbreak in each of the 16 towns when the outbreak starts at a random location in the county, or when it starts in one of eight different towns: Cloone, Dromahair, Fenagh, Kinlough, Leitrim, Manorhamilton, Mohill or Tullaghan. Each column represents a different starting location of the outbreak. The eight towns are selected to get a range of centralities but were also chosen using the town similarities data discussed in the previous section so that any differences found can be more easily attributed to the differences in centrality.

The results in Table 8 show a range of percents across the different towns. Some towns appear to be more stable than others. For example, Leitrim where the percent of runs leading to an outbreak ranges from 14 to 22 depending on where the outbreak starts, while Dromod ranges from 7.7 to 27.7. One thing to notice with the towns is that the towns that tend to have a more stable percentage change of outbreak despite the starting location are those that tend to have lower centrality scores across all four measures. Similarly, the towns with the lowest average percentage across all starting locations have lower degree centralities. Table 9 shows the Pearson correlations between the different centralities of the towns and the range of percentage of runs that lead to outbreaks for each town based on starting location and the average percentage of runs that lead to an outbreak across all starting locations.

To analyze correlations we use the following scale: 0 corresponds to no linear relationship, 0 to 0.3 or 0 to −0.3 corresponds to a weak linear relationship, 0.3 to 0.7 or −0.3 to −0.7 corresponds to a moderate relationship and 0.7 to 1.0 or −0.7 to −1.0 corresponds to a strong linear relationship [22]. From the table it can be seen that there is a moderate relationship between total degree centrality and in-degree centrality and both the range and mean values. Closeness centrality also has a negative moderate relationship with the mean value a weak relationship with range. From these results it is clear that there is a relationship between centrality and how susceptible a town is to an outbreak spreading to it in a network of other towns. In the next sections we aim to look at the differences in the results by town and how the centrality of the town influences those difference.

### 3.1. Degree Centrality

The results listed in Table 9 indicate that total degree centrality is moderately correlated with both the average percent of runs leading to an outbreak across all starting locations and the range of percent of runs leading to an outbreak as well. To look deeper into the relationship between total degree centrality and the results across the towns the correlations between the percent of runs leading to an outbreak in a given town and the total degree centrality of that town are found across all starting locations of the outbreak. These correlations are presented in Table 10.

The results show that for all infection starting locations except for Manorhamilton there is a moderate correlation between the centrality of a town and the percent of runs that lead to an outbreak in that town. This indicates that towns that are more connected are more likely to experience an outbreak. For outbreaks that start in Manorhamilton there is a weak negative correlation between the centrality of a town and the percent of runs that lead to an outbreak in that town. Because the correlation is weak we do not consider the negative part of the correlation as important and instead focus on the weak piece of the correlation. From Table 7 we see that Manorhamilton is the town with the highest total degree centrality, therefore, we can interpret these results as showing that when an outbreak begins in a town with a very high total degree centrality, then the centrality of other towns is less important to how an infectious disease spreads. This can be further emphasized when looking at the town with the next lowest correlation, Mohill, which is also the town with the second highest total degree centrality. However, if an outbreak starts in a town that has a lower total degree centrality, the total degree centrality of the other towns has a larger impact on the progress of the infection.

Because there are multiple parts of total degree centrality, it is made up of both agents commuting into a town and agents commuting out of a town, we break total degree centrality down further into in-degree centrality and out-degree centrality. This will allow us to understand better what has an influence on an outbreak, agents coming in or going out or if they are both equally important.

#### 3.1.1. In-Degree Centrality

Of the different measures of centrality, in-degree centrality seems to be the most highly correlated with whether an outbreak will spread to a given town. Logically, this makes sense as an outbreak can only spread to a town if there are agents commuting into the town. Table 11 shows the correlations between the percent chance of an outbreak in a given town and the in-degree centrality of the town broken down by the starting location of the outbreak.

The correlations show that for all starting locations except for Manorhamilton and Mohill the in-degree centrality has a moderate relationship with the percent of runs leading to an outbreak. Again Manorhamilton has a negative correlation but it is very weak, almost 0. Thus, we do not think that the negative correlation is signficant. Both Manorhamilton and Mohill have high in-degree centralities while Cloone, Dromahair, Fenagh, Kinlough, Leitrim and Tullaghan have lower in-degree centralities. This is similar to the results seen in the total degree centrality and thus can be interpreted in a similar way: when an outbreak starts in a town with high in-degree centrality the in-degree centrality of another town in the network does not have as large of an effect on whether the outbreak will spread to that other town compared to when an outbreak starts in a town with lower in-degree centrality.

#### 3.1.2. Out-Degree Centrality

From the results in Table 9 it can be seen that out-degree centrality only has a very weak relationship with both the range and mean value of percent of runs that lead to an outbreak. Similarly, if we look at the correlations between the percent of runs that lead to an outbreak and the out-degree centrality of the town by starting location of the outbreak in Table 12 we can see that if the initial outbreak starts in Fenagh, Leitrim, or Mohill there is a moderate relationship between the out-degree centrality of other towns in the network and the percent of runs that lead to an outbreak in that town, but all other towns have a weak correlation.

Looking further into the results we can see that out-degree centrality might have an effect that is not captured in the correlations. For example, when an outbreak starts in a town with high out-degree centrality in general the percent of runs that lead to an outbreak in the other towns tend to be higher than when an outbreak starts in a town with low out-degree centrality. Leitrim and Mohill are two towns that are similar in most of their attributes (with a low euclidean distance between them as can be seen in Figure 1) but Leitrim has a high out-degree centrality while Mohill has a significantly lower out-degree centrality. When the outbreak starts in Leitrim, the average percent of runs that lead to an outbreak in the other towns in the model is 21.7 and it is 15.8 when the outbreak starts in Mohill. Comparing the results town to town shows that for almost every town, they have higher percentage of runs that lead to an outbreak when the outbreak starts in Leitrim versus Mohill. This seems to make sense as a higher out-degree centrality would mean that more agents from that town are commuting to other towns resulting in a higher percentage of runs that lead to an outbreak spreading.

### 3.2. Closeness Centrality

Closeness centrality gives a measure of how close a town is to every other town in the network. For this paper we focuses on how close in geographic distance the town is to the other towns in the network for closeness centrality as we feel that geographic distance should be an important factor in the spread of a disease: an outbreak should be more likely to spread to a town that is close to several other towns than a secluded town in the network as there will be more movement between towns in the former case. Looking at Table 9 it can be seen that there is a moderate relationship between the closeness centrality of a town and the average percent of runs that lead to an outbreak across all starting locations. Looking into this farther, the correlations between the percent of runs that lead to an outbreak and the closeness centrality of that town for each starting location of the outbreak can be seen in Table 13.

The table shows that two starting locations, Manorhamilton and Tullaghan, lead to a moderate negative relationship between the closeness centrality of the town and if the outbreak will spread there. Two additional towns, Dromahair and Kinlough show a weak to moderate negative relationship. These four towns have the highest closeness centrality, are the farthest away from the majority of towns in the model, of the eight towns that are studied as starting locations of the model and are four of the five highest closeness centralities across the sixteen towns studied. From this we can make an assumption that the less connected the town where an outbreak starts is when considering closeness centrality, the more important the closeness centrality of the other towns the outbreak spreads to is. This pattern occurs regardless of the other values for degree centrality, Manorhamilton and Tullaghan both have the highest closeness centrality scores and similar correlations in Table 13 but have different values for total, in, and out-degree centrality. Manorhamilton has the highest total and in-degree centrality and a lower out-degree centrality while Tullaghan has the lowest total, in, and out-degree centrality of all the towns.

As the towns in the simulation have many other characteristics that could influence the spread of an outbreak, such as population size and age structure, to determine if the relationship between the closeness centrality of the town where the outbreak starts and where the outbreak spreads exists regardless of the other characteristics of the town we look at the euclidean distance between towns from Figure 1 along with the closeness centrality and model results. Manorhamilton and Dromahair can be seen as two towns with a closer euclidean distance in Figure 1 and the pattern occurs in both towns, while both Tullaghan and Kinlough have farther euclidean distance to Manorhamilton and again the pattern occurs in all three towns. We believe that this shows the other characteristics of the towns do not influence the result that the less connected the town where the outbreak starts is when considering closeness centrality, the more important the closeness centrality of the other towns the outbreak spreads to is.

The negative correlation here is interpreted as a higher closeness centrality results in a lower percentage of runs that lead to an outbreak and vice versa. Although an obvious explanation for this is difficult to determine, it would seem that the closer to other towns a given town is it should be more susceptible to an outbreak not less, this result could be down to interactions with other factors and the towns with moderate correlations that are further discussed in the next section.

### 3.3. Distance and Centrality

Closeness centrality gives a measure of how close a town is to every other town in the network: is the town located near a lot of other towns or is it more secluded. However, a factor that is associated with closeness centrality but not quite captured in it is the distance between the town where the outbreak initially starts and the town where it spreads. For example, if the outbreak starts in Mohill, it is only approximately 12 km to Fenagh but 80 km to Kinlough. It would thus seem that it should be more likely that an outbreak would spread from Mohill to Fenagh than Mohill to Kinlough. Evidence for this can be seen if we look at the results in Table 8 and the approximate distances between towns. Table 14 provides the approximate distances between towns with the percent of runs that lead to an outbreak for comparison.

When comparing Table 8 and Table 14 it can be seen that for some cases distance to the town does have an effect on the results. When an outbreak starts in Tullaghan, a town with high closeness centrality and low degree centrality, 41% of runs lead to an outbreak in Manorhamilton. Manorhamilton is a town that is close in distance to Tullaghan, approximately 28 km, and is also a town that has high degree centrality. This is the highest percent of runs that lead to an outbreak for Manorhamilton across all considered starting locations of the outbreak. The next four towns with the highest percent of runs that lead to an outbreak when it begins in Tullaghan are Dromahair (31.3%), Kinlough (31.3%), Mohill (25%), and Lurganboy (24.3%). Dromahair, Kinlough, Lurganboy and Manorhamilton are the four towns closest to Tullaghan that are considered in the model and Mohill has the second highest total and in-degree centrality of all of the towns. From this we can consider that when an outbreak starts in a town with low centrality an outbreak is likely to spread to the towns that are both nearby and with those towns with high degree centrality.

Looking across the table it can be seen that for several towns the lowest percent of runs leading to an outbreak come when the outbreak starts in a location that is far away. For example, in Carrigallen the lowest four values for percent of runs leading to an outbreak come when the outbreak starts in Dromahair, Kinlough, Manorhamilton and Tullaghan. These towns are all farther away from Carrigallen then any other town considered in the model. The distance seems to counteract the high in-degree centrality for Manorhamilton and the relatively high out-degree centrality in Dromahair and protects Carrigallen from an outbreak occurring. A similar phenomenon happens for the town of Dromod where the percent of runs leading to an outbreak when the initial outbreak starts in Kinlough, Manorhamilton or Tullaghan are significantly lower than when the outbreak starts in other towns. There are also cases of the reverse where towns such as Drumsna and Lurganboy show higher percents when the town the initial outbreak starts in a close town. Lurganboy is very close to Manorhamilton and if the outbreak starts in Manorhamilton there is a 27% chance that the outbreak will spread to Lurganboy. However, if the outbreak starts in Mohill, a town that is similar to Manorhamilton in both euclidean distance and degree centrality an outbreak starts in Lurganboy only 7.3% of the time. The short distance to Manorhamilton clearly plays a key role in the susceptibility of Lurganboy to an outbreak.

Distances might also play a role in the moderate negative correlations found with closeness centrality in Table 13. The four towns with a moderate correlation, (Manorhamilton, Dromahair, Kinlough and Tullaghan) all have lower closeness centralities but are close to each other and farther away from the other towns. These moderate negative correlations might be capturing the effects of distance between towns. If a town has a low closeness centrality, the distance to the other towns might be more important to whether an outbreak spreads to the other town than the closeness centrality of the other towns.

## 4. Discussion

Being able to scale up a town agent-based model to model a network of towns within a region is important in being able to capture and understand the spread of an infectious disease. No town exists in isolation and movements into and out of a town can greatly influence the susceptibility of a town to an outbreak. In this article, we have shown that assumptions can be made to scale an agent-based model from a single town model to a regional model. These assumptions and changes have been shown to not significantly influence the results of the model and allow us to better model a county or region. The scaled up model allowed us to study more than just factors affecting a single town but how the centrality of a town and how connected it was to other towns in the region influences an outbreak in a town. Agent-based models are particularly suited to this task as they capture the interactions of different factors and as we have shown it is not just one type of centrality that effects whether an outbreak spreads to a town but multiple types of connectedness as well as other information about the network such as the town where the initial outbreak occurred along with the other factors that were identified in Hunter et al. [3].

Modelling how agents movements influence the course of an outbreak is important in studying how to react when an outbreak occurs. If an outbreak starts in a given region, towns that are more susceptible in that region can be a focus of the response, with more resources sent to these towns. Our results showed that compared to the correlations with in-degree centrality the correlations for out-degree centrality are markedly different than those with total degree centrality. One possible conclusion for this is that for the spread of an infectious disease through a network agents commuting into a town are more important than agents commuting out of a town. Therefore, the planning of an intervention to stop the spread of a disease to a town should focus on who commutes into a town versus those who commute out. In addition, the location of the initial cases can be used to guide responses. For example, if the outbreak starts in a town with high in-degree centrality closing the town might prevent further spread of an outbreak while if the town has low degree centrality the best course of action might be closing things such as schools or workplaces in nearby towns with high degree centrality as this would stop the outbreak from spreading into the high degree centrality town and then to many other locations from there. Additional simulations could be run including such restrictions to better understand how they could help to reduce the size and severity of an outbreak.

Although we only model a single county it would be realistic to use the same method to model multiple counties or an entire country. Data, and in this case open data, is particularly important because using data allows us to create a realistic agent-based model for the spread of infectious diseases that can be used to help determine the factors that lead to the susceptibility of a given town. When scaling up the model it is important to know what data will make a difference in the results and what data will not. For example, we were able to reduce the environmental fidelity of the model by not including as much GIS data such as zoning data without overly impacting the results. Additional data were used in other areas such as creating a more realistic transportation model.

Additionally, even though our model simulates the spread of measles through a region in Ireland, one advantage of using an agent-based model is that it is easily adaptable to other diseases. By changing the basic disease parameters such as the infectivity rate, the incubation period or the infectious period we can model a disease with different dynamics. Adjusting the parameters can also be done to look at how an outbreak might differ when an exact value for the parameters is not known. For instance, changing the infectious period of the disease. While this can also be done easily with other types of infectious disease models such as an equation-based SEIR model, the agent-based model allows us to easily add other factors and agent actions. For example, with COVID-19 many cases are mild while a few are more severe. Especially early on in the pandemic, agents’ actions might differ if they have a mild case versus a severe case. Those with more mild cases might be less likely to stay home and will go out and infect more individuals while more severe cases will be unable to go out and will only be able to infect those close to them or health care workers. An agent-based model can easily make adjustments for different groups of people and with an existing model for a population will help prepare for future outbreaks and pandemics by allowing us to learn as much as we can about an outbreak before it occurs.

## Figures and Tables

**Figure 1 ijerph-17-03119-f001:**
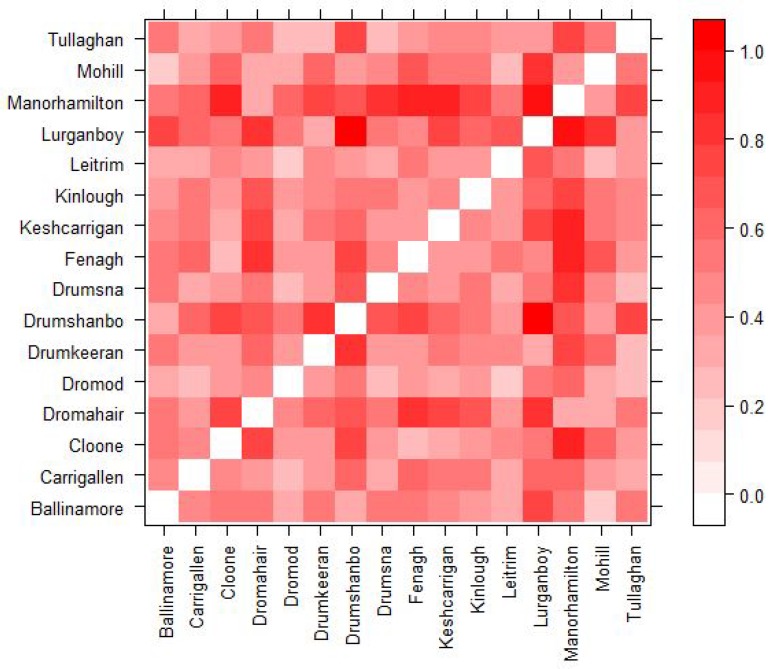
Distance Matrix showing the normalized euclidean distance between towns.

**Table 1 ijerph-17-03119-t001:** Percent of runs leading to an outbreak for the detailed environment model and the reduced fidelity environment model.

Town	Detailed Environment	Reduced Environment
Schull	70.3 (65.2, 75.5)	71.0 (65.9, 76.1)
Tramore	72.0 (66.9, 77.1)	65.3 (59.9, 70.7)

**Table 2 ijerph-17-03119-t002:** Percent of runs leading to an outbreak for a steady versus a varying initial population.

Town	Varying Initial Population	Constant Initial Population
Schull	70.3 (65.2, 75.5)	65 (59.6, 70.4)
Tramore	72.0 (66.9, 77.1)	69.3 (64.1, 74.6)

**Table 3 ijerph-17-03119-t003:** Summary statistics for the number of agents infected per run for a steady versus a varying initial population.

Town	Schull Varying	Schull Constant	Tramore Varying	Tramore Constant
Min	1.00	1.00	1.00	1.00
1st Quartile	1.00	1.00	1.00	1.00
Median	5.00	3.00	11.00	5.00
Mean	10.22	5.27	27.73	21.39
3rd Quartile	17.00	8.00	47.25	36
Max	63.00	29.00	129.00	217.00
Standard Deviation	11.26	5.29	33.89	29.51

**Table 4 ijerph-17-03119-t004:** Differences between the Hunter et al. [3] town model and the scaled up region model by component of the model.

Component	Town Model [3]	Region Model
Environment	Small areas are made up of multiple patches	Small areas are made up of single patch
Society	Model creates new society each run	Model loads a constant society each run
Transportation	Random movements through the town	Movement based on gravity model
Disease	No Change	No Change

**Table 5 ijerph-17-03119-t005:** Results for the town models and the county model.

Town	Town Model	Town Model with Commuting	County Model
Manorhamilton	66.3 (61.0, 71.7)	41.0 (35.4, 46.6)	52.7 (47.0, 58.3)
Kinlough	48.0 (42.3, 53.7)	32.0 (26.7, 27.3)	42.5 (36.1, 47.2)

**Table 6 ijerph-17-03119-t006:** Town Characteristics.

Town	Population	Area (km^2^)	Small Areas	Percent Students
Ballinamore	1271	18.22	7	45.9
Carrigallen	953	29.75	5	43.4
Cloone	405	18.03	2	47.9
Dromahair	1615	34.33	9	42.4
Dromod	971	22.16	5	44.3
Drumkeeran	531	28.73	3	46.3
Drumshanbo	1594	15.95	8	46.4
Drumsna	760	21.75	3	43.0
Fenagh	487	15.40	2	48.0
Keshcarrigan	551	10.43	2	45.2
Kinlough	1141	22.26	5	49.6
Leitrim	1298	22.90	6	44.4
Lurganboy	497	29.62	2	46.3
Manorhamilton	2106	33.63	11	43.9
Mohill	1395	22.34	8	43.9
Tullaghan	793	25.43	4	44.5

**Table 7 ijerph-17-03119-t007:** Normalized centrality by town.

Town	Total Degree	In-Degree	Out-Degree	Closeness
Ballinamore	0.67	0.45	0.91	0.95
Carrigallen	0.47	0.45	0.23	0.49
Cloone	0.16	0.09	0.27	0.64
Dromahair	0.32	0.17	0.51	0.44
Dromod	0.37	0.21	0.63	0.53
Drumkeeran	0.34	0.29	0.27	0.75
Drumshanbo	0.89	0.76	0.70	0.95
Drumsna	0.27	0.04	0.74	0.81
Fenagh	0.26	0.17	0.38	0.96
Keshcarrigan	0.33	0.26	0.28	1.00
Kinlough	0.18	0.18	0.01	0.07
Leitrim	0.51	0.22	1.00	0.89
Lurganboy	0.19	0.02	0.56	0.38
Manorhamilton	1.00	1.00	0.24	0.40
Mohill	0.90	0.87	0.41	0.71
Tullaghan	0.00	0.00	0.00	0.00

**Table 8 ijerph-17-03119-t008:** Percent of runs leading to an outbreak in each of the 16 towns when the outbreak starts in a random location or one of the eight selected towns.

	Random Start	Cloone	Dromahair	Fenagh	Kinlough	Leitrim	Manorhamilton	Mohill	Tullaghan
Ballinamore	24.0	24.7	20.3	31.3	17.7	29.3	15.7	24.0	19.0
Carrigallen	17.3	25.0	11.3	22.3	13.3	21.7	7.7	16.3	11.0
Cloone	7.7	-	5.3	10.0	6.0	12.3	5.3	15.3	8.7
Dromahair	18.3	19.0	-	23.7	26.7	28.3	31.0	19.3	31.3
Dromod	18.3	27.0	15.3	19.3	13.0	27.7	7.7	22.0	13.0
Drumkeeran	7.3	9.3	14.7	8.3	13.7	8.7	15.0	7.3	10.7
Drumshanbo	18.6	17.7	22.7	23.3	23.0	29.7	18.7	18.0	15.3
Drumsna	14.3	19.7	13.7	12.3	12.0	18.3	8.0	20.3	9.7
Fenagh	11.7	15.3	10.3	-	13.3	15.7	7.3	10.3	8.0
Keshcarrigan	9.3	5.7	10.7	13.7	9.0	16.7	6.0	6.3	8.0
Kinlough	21.0	23.0	24.3	20.3	-	22.3	30.7	17.7	31.3
Leitrim	18.0	19.0	19.3	22.0	16.7	-	16.0	18.7	14.3
Lurganboy	6.7	8.0	14.3	11.3	13.3	18.3	27.0	7.3	24.3
Manorhamilton	22.7	25.0	34.3	24.7	27.3	28.3	-	18.0	40.7
Mohill	24.3	36.7	19.3	27.3	24.3	28.3	16.7	-	25.0
Tullaghan	15.3	18.0	18.0	16.7	20.0	20.0	26.0	16.0	-

**Table 9 ijerph-17-03119-t009:** Correlations between centrality and the range and average percentage outbreaks.

Centrality	Range	Mean
Total Degree	0.51	0.58
In-Degree	0.53	0.58
Out-Degree	0.02	0.07
Closeness	−0.29	−0.38

**Table 10 ijerph-17-03119-t010:** Correlations between the percent of runs that lead to an outbreak and the total degree centrality of the town by starting location of the outbreak.

Start of Outbreak	Correlation
Random Start	0.64
Cloone	0.51
Dromahair	0.62
Fenagh	0.68
Kinlough	0.62
Leitrim	0.65
Manorhamilton	−0.12
Mohill	0.34
Tullaghan	0.40

**Table 11 ijerph-17-03119-t011:** Correlations between the percent of runs that lead to an outbreak in a town and the in-degree centrality of the town by starting location of the outbreak.

Start of Outbreak	Correlation
Random Start	0.61
Cloone	0.52
Dromahair	0.64
Fenagh	0.62
Kinlough	0.62
Leitrim	0.57
Manorhamilton	−0.08
Mohill	0.22
Tullaghan	0.44

**Table 12 ijerph-17-03119-t012:** Correlations between the percent of runs that lead to an outbreak and the out-degree centrality of the town by starting location of the outbreak.

Start of Outbreak	Correlation
Random Start	0.19
Cloone	0.07
Dromahair	0.02
Fenagh	0.29
Kinlough	0.03
Leitrim	0.39
Manorhamilton	−0.16
Mohill	0.41
Tullaghan	−0.20

**Table 13 ijerph-17-03119-t013:** Correlations between the percent of runs that lead to an outbreak and the closeness centrality of the town by starting location of the outbreak.

Start of Outbreak	Correlation
Random Start	−0.11
Cloone	−0.17
Dromahair	−0.28
Fenagh	0.07
Kinlough	−0.27
Leitrim	−0.08
Manorhamilton	−0.63
Mohill	−0.10
Tullaghan	−0.65

**Table 14 ijerph-17-03119-t014:** Approximate distances between each town in km and the percent of runs that lead to an outbreak.

	Cloone	Dromahair	Fenagh	Kinlough	Leitrim	Manorhamilton	Mohill	Tullaghan
Ballinamore	12 *(24.7)*	51 *(20.3)*	5 *(31.3)*	77 *(17.7)*	28 *(29.3)*	56 *(15.7)*	17 *(24.0)*	82 *(19.0)*
Carrigallen	14 *(25.0)*	66 *(11.3)*	16 *(22.3)*	89 *(13.3)*	34 *(21.7)*	71 *(7.7)*	19 *(16.3)*	93 *(11.0)*
Cloone	-	55 *(5.3)*	10 *(10.0)*	81 *(6.0)*	25 *(12.3)*	60 *(5.3)*	8 *(15.3)*	88 *(8.7)*
Dromahair	55 *(19.0)*	-	50 *(23.7)*	32 *(26.7)*	37 *(28.3)*	14 *(31.0)*	53 *(19.3)*	39 *(31.3)*
Dromod	17 *(27.0)*	56 *(15.3)*	21 *(19.3)*	82 *(13.0)*	36 *(27.7)*	61 *(7.7)*	9 *(22.0)*	89 *(13.0)*
Drumkeeran	40 *(7.3)*	15 *(14.7)*	35 *(8.3)*	41 *(13.7)*	23 *(8.7)*	20 *(15.0)*	39 *(7.3)*	48 *(10.7)*
Drumshanbo	22 *(17.7)*	32 *(22.7)*	17 *(23.3)*	59 *(23.0)*	7 *(29.7)*	37 *(18.7)*	21 *(18.0)*	65 *(15.3)*
Drumsna	18 *(19.7)*	47 *(13.7)*	21 *(12.3)*	73 *(12.0)*	10 *(18.3)*	53 *(8.0)*	11 *(20.3)*	80 *(9.7)*
Fenagh	10 *(15.3)*	50 *(10.3)*	-	77 *(13.3)*	18 *(15.7)*	56 *(7.3)*	12 *(10.3)*	84 *(8.0)*
Keshcarrigan	14 *(5.7)*	41 *(10.7)*	8 *(13.7)*	68 *(9.0)*	10 *(16.7)*	46 *(6.0)*	13 *(6.3)*	74 *(8.0)*
Kinlough	81 *(23.0)*	32 *(24.3)*	77 *(20.3)*	-	63 *(22.3)*	21 *(30.7)*	78 *(17.70*	7 *(31.3)*
Leitrim	25 *(19.0)*	37 *(19.3)*	18 *(22.0)*	63 *(16.7)*	-	49 *(16.0)*	29 *(18.7)*	70 *(14.3)*
Lurganboy	62 *8.0)*	13 *(14.3)*	58 *(11.3)*	20 *(13.3)*	45 *(18.3)*	3 *(27.0)*	61 *(7.3)*	27 *(24.3)*
Manorhamilton	60 *(25.0)*	14 *(34.3)*	56 *(24.7)*	21 *(27.3)*	49 *28.3)*	-	58 *(18.0)*	28 *(40.7)*
Mohill	8 *(36.7)*	53 *(19.3)*	12 *(27.3)*	78 *(24.3)*	20 *(28.3)*	58 *(16.7)*	-	90 *(25.0)*
Tullaghan	88 *(18.0)*	39 *(18.0)*	84 *(16.7)*	7 *(20.0)*	70 *(20.0)*	28 *(26.0)*	90 *(16.0)*	-

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
