# Peer review of "A Model for the Spread of Infectious Diseases in a Region"

_ijerph, 2020, doi:10.3390/ijerph17093119_

Round 1

Reviewer 1 Report

Line 224, authors calculated that an infected person remains infectious during 8 days, but there is some data that calculate this period at least 14 days (in some cases this period is near 21 days). Please employ a viral sheeding period of 14 days instead of 8 days and show the new data. 

Why in Manorhamilton had a negative correlation in table 9? The explanation in lines 391-395 is not enough. The same phenomena is showed in table 10 and 11, but not in table 12. Provide a more wide explanation about the behavior of Manorhamilton. 

Author Response

We would like to thank reviewer 1 for their thoughtful and insightful comments on our paper.  The comments allowed us to improve our work and we have integrated their suggestions in to our revised version of the paper. Please find below our point by point response discussing how we addressed each of the referees comments. They are separated by reviewer. Our responses are in italics.

Sincerely,

Elizabeth Hunter

Reviewer 1

 Line 224, authors calculated that an infected person remains infectious during 8 days, but there is some data that calculate this period at least 14 days (in some cases this period is near 21 days). Please employ a viral sheeding period of 14 days instead of 8 days and show the new data. 

We have cited our source for using 8 days as the infectious period but also use a distribution so some agents will have an infectious period of more than 8 days.

As the focus of the article is on scaling up a model that has been previously published from a town level to a county level we do not think it would be appropriate to change the disease parameters as this would make the results between the two models incomparable and would not be part of the process of scaling up the model from a town to a county.

However, we added the following text to our future work section to discuss the possibility of looking at the results of the model using different viral shedding periods. Starting at line 611 with the new text beginning at line 614:

“Additionally, even though our model simulates the spread of measles through a region in Ireland, one advantage of using an agent-based model is that it is easily adaptable to other diseases.  By changing the basic disease parameters such as the infectivity rate, the incubation period or the infectious period we can model a disease with different dynamics. Adjusting the parameters can also be done to look at how an outbreak might differ when an exact value for the parameters is not known. For instance, changing the infectious period of the disease. While this can also be done easily with other types of infectious disease models such as an equation based SEIR model, the agent-based model allows us to easily add other factors and agent actions.  For example, with COVID-19 many cases are mild while a few are more severe. Especially early on in the pandemic agents actions might differ if they have a mild case versus a severe case. Those with more mild cases might be less likely to stay home and will go out and infect more individuals while more severe cases will be unable to go out and will only be able to infect those close to them or health care workers. An agent-based model can easily make adjustments for different groups of people and having an existing model for a population will help prepare for future outbreaks and pandemics by allowing us to learn as much as we can about an outbreak before it occurs. 

Why in Manorhamilton had a negative correlation in table 9? The explanation in lines 391-395 is not enough. The same phenomena is showed in table 10 and 11, but not in table 12. Provide a more wide explanation about the behavior of Manorhamilton. 

Manorhamilton has a negative correlation in tables 9, 10, 11 and 12.  In tables 9, 10 and 11 it is a negative but weak correlation. The text that that was originally included concerning the negative correlation in table 9 was this:

“Manorhamilton is the town with the highest total degree centrality, therefore, we can interpret these results as the higher the total degree centrality of the town the outbreak begins in the less important the total degree centrality of the other towns is to the spread of the infectious disease.”

To further clarify the negative correlation we included added to that text by including the following starting on line 439 with the new text starting on line 441:

“The results show that for all infection starting locations except for Manorhamilton there is a moderate correlation between the centrality of a town and the the percent of runs that lead to an outbreak in that town. This indicates that towns that are more connected are more likely to experience an outbreak. For outbreaks that start in Manorhamilton there is a weak negative correlation between the centrality of a town and the percent of runs that lead to an outbreak in that town.  Because the correlation is weak we do not consider the negative part of the correlation as important and instead focus on the weak piece of the correlation From Table 7 we see that Manorhamilton is the town with the highest total degree centrality, therefore, we can interpret these results as showing that when an outbreak begins in a town with a very high total degree centrality, then the centrality of other towns is less important to how an infectious disease spreads. This can be further emphasized when looking at the town with the next lowest correlation, Mohill, which is also the town with the second highest total degree centrality.   However, if an outbreak starts in a town that has a lower total degree centrality, the total degree centrality of the other towns has a larger impact on the progress of the infection.” 

Additionally in the in-degree centrality section we included  text mentioning the weak negative correlation for Manorhamilton. Starting at line 462 with the new text beginning on line 463:   

“The correlations show that for all starting locations except for Manorhamilton and Mohill the in-degree centrality has a moderate relationship with the percent of runs leading to an outbreak. Again Manorhamilton has a negative correlation but it is very weak, almost 0. Thus we do not think that the negative correlation is significant. Both Manorhamilton and Mohill have high in-degree centralities while Cloone, Dromahair, Fenagh, Kinlough, Leitrim and Tullaghan have lower in-degree centralities.  This is similar to the results seen in the total degree centrality and thus can be interpreted in a similar way: when an outbreak starts in a town with high in-degree centrality the in-degree centrality of another town in the network does not have as large of an effect on whether the outbreak will spread to that other town compared to when an outbreak starts in a town with lower in-degree centrality.”

And we added the following text in the out degree centrality section. Starting at line 472 with the new text on line 478:

“From the results in Table 11 it can be seen that out degree centrality only has a very weak relationship with both the range and mean value of percent of runs that lead to an outbreak. Similarly, if we look at the correlations between the percent of runs that lead to an outbreak and the out degree centrality of the town by starting location of the outbreak in Table 11 we can see that if the initial outbreak starts in Fenagh, Leitrim, or Mohill there is a moderate relationship between the out degree centrality of other towns in the network and the percent of runs that lead to an outbreak in that town, but all other towns have a weak correlation.”

Reviewer 2 Report

Generally throughout, I'd be more liberal with commas. This may be a difference in style, but some sentences are unnecessarily difficult to parse without commas to break them up.

I'll be absolutely honest - I'm stuck at home with an ancient laptop, and the antialiasing is turned off, and some of the line numbers below may be wrong. I apologize in advance.

16: routines, it

17: prepare and be ready may be redundant

18: lower-than-desired

18: mean that outbreaks

24: measles_[1]

24: population, infectious

43: larger-scale model

57: one advantage <...> is

58: agents' actions?

60: The citation here has an odd = in it.

87: openly-available

89: among

100: It's helpful to know that this is a well-defined term in the Irish census, but it may be useful to be told in-line that this is a formal term. Something like "so-called," or use italics as below on 105.

109: It may be useful to recall somewhere here or above that this is a measles model, conditioned and validated on real data, perhaps on 87. Give us a sentence or two to establish the model in the mind before we tinker with it.

118: significantly_-_the

118: might just note that the confidence intervals overlap. "results" is ambiguous

120: This definition might be moved somewhere above, so that we're more certain of it before we start tinkering with it.

123: are

138: Do these also come from the CSO? These data are rare and will influence the model outcome, so the reader will naturally be curious about their origin.

142: probably replace "society" on 110 with synthetic population, as here.

148: agents' or agent, no ,

151: influence our results, holding

152: by the end of this paragraph, I'm not sure which one you're doing. You clarify in the next, but it might be worth just saying that here, or starting the next paragraph with hence or something. I'm left with a question and then have to figure out that the next paragraph answers it.

162: Generally, I would expect Tables 1 and 2 and 3 to belong to a results section. If not, if these are used to justify choices later, then there need to be some sentences in place saying something like: we see no difference between the static and resimulated populations, so in the final analysis we'll ______. In the middle of reading this, I cannot tell what's a final result and what's a step along the path.

175-187: I'm confused by this. It seems that there are still subdivisions within the small areas (e.g. home vs work) and so they are not in the same physical location. It seems you've reduced the number of states within the small areas, but not completely - there is still the possibility of movement from discrete state to discrete state. Sure, you have reduced the dimension of the coordinates in the small area, but not so completely that they're all in the same location.

220: Once recovered,

224: Once infectious,

236: Scaled-up Model

256: Feels like results again.

294: or region, we

358: This is my major difficulty reading this paper: "for each set of runs." Which runs? I've seen a variety of simulations above, some with more influential choices than others, and the casual reader at this point will not be sure precisely which model we're using to make inferences.

The structure is clear-ish but leaves a lot out. I need to know: which choices do we have to make when getting to the final model, and which choices have we actually made, and why? It's fine if you put the various tables above in the methods section to justify the various simplifications you've made, but the first paragraph of the results section needs to recall all of this and list what was done where. I need to know precisely what the final model is that is being run and analyzed.

It may be useful to have a table describing the orthodox Hunter [3] model and the version used here. The reader will lose track of what has been changed, after all the effort you put into demonstrating that the changes were feasible. Let me know what you're changing, why, why it's alright (your simulations) and then recap.

404 and elsewhere: in-degree

418: same

438: Conclusions such as this are probably best moved to a discussion.

The vast majority of the good and necessary work has already been done, but pieces of the paper need to be shuttled around. I'm perfectly happy to see tables of results in the methods, as long as it's been made clear that these intermediate simulations are used to justify the final thing, the results of which only appear in the results. I'm stupid - tell me if this table is what I'm really here for, or if this is an intermediate step.

Really make sure that every number and interpretation is where it belongs. The flow is clear if you go back and forth, but the average reader will likely not.

Author Response

We would like to thank the reviewers for their thoughtful and insightful comments on our paper.  The comments allowed us to improve our work and we have integrated their suggestions in to our revised version of the paper. Please find below our point by point response discussing how we addressed each of the referees comments. They are separated by reviewer. Our responses are in italics. 

Sincerely,

Elizabeth Hunter

Reviewer 2

Generally throughout, I'd be more liberal with commas. This may be a difference in style, but some sentences are unnecessarily difficult to parse without commas to break them up.

I'll be absolutely honest - I'm stuck at home with an ancient laptop, and the antialiasing is turned off, and some of the line numbers below may be wrong. I apologize in advance.

16: routines, it

Changed the text to include the comma

17: prepare and be ready may be redundant

Got rid of “be ready” in the text

18: lower-than-desired

Changed the text

18: mean that outbreaks

Changed the text

24: measles_[1]

Added the extra space

24: population, infectious

Added the comma

43: larger-scale model

Added the dash

57: one advantage <...> is

Changed “are” to “is”

58: agents' actions?

Added the apostrophe

60: The citation here has an odd = in it.

Fixed the citation

87: openly-available

Added the dash

89: among

Changed “between” to “among”

100: It's helpful to know that this is a well-defined term in the Irish census, but it may be useful to be told in-line that this is a formal term. Something like "so-called," or use italics as below on 105.

On line 121 we put “small areas” in italics and in lines 122 – 123 moved the definition “The smallest geographic area that the Irish census is aggregated over. Each small area contains between 50 to 200 dwellings” From the footnote to the text. 

109: It may be useful to recall somewhere here or above that this is a measles model, conditioned and validated on real data, perhaps on 87. Give us a sentence or two to establish the model in the mind before we tinker with it.

We have added text in the introductory paragraph of the methodology section to include more information about the original model and its validation.  Starting at line 87:

“The Hunter et al. [3] model uses openly-available data to model towns in Ireland. The model is an agent-based model that was designed to simulate the spread of measles through an Irish town, however, the towns are considered in isolation with no commuting between towns. The model was validated in a number of ways: comparing the basic disease dynamics to expected disease dynamics, doing a sensitivity analysis on a number of key parameters and comparing the results to real data from measles outbreaks in Ireland. In this work, we focus on scaling up this model, from modelling towns in isolation, to take into account the interactions among populations in different towns within a region.”

118: significantly_-_the

Added spaces

118: might just note that the confidence intervals overlap. "results" is ambiguous

On lines 138-139 we changed  “The results for the detailed environment are within the confidence intervals for the reduced environment and vice versa.”

to

“the confidence intervals for the detailed environment and the reduced environment overlap”

So the paragraph discussing the results for the environmental component validation is now (starting at line 132):

To test the effect of reducing the environmental fidelity of the model on the model’s output, we run the model for two towns Schull and Tramore 300 times with the original environment and 300 times with the reduced environmental fidelity. All other parameters are held constant and the results are compared.  Table 1 shows the percent of the 300 runs that lead to a measles outbreak for both Schull and Tramore and the 95%confidence intervals for both. From the results we can see that the abstraction of small areas to points did not cause the results between the two versions of the towns to vary significantly - the confidence intervals for the detailed environment and the reduced environment overlap. From this result we argue that we can use the reduced environmental version of the model without significantly altering the model

120: This definition might be moved somewhere above, so that we're more certain of it before we start tinkering with it.

We moved the definition of small areas into the text (see response to comment on line 100) and adjusted the text in line 143 to reference this “As mentioned in the previous section, small areas are geographic census areas that contain between 50 to 200 dwellings.”

The new paragraph starting on line 142 reads

“The society of the model is created using Irish Census data from the CSO [12]. The CSO data is at the small area level. As mentioned in the previous section, small areas are geographic census areas that contain between 50 to 200 dwellings. They are the smallest area over which the Irish census data is aggregated. For each small area we create a population that reflects the population statistics of that small area including age, sex, household size and economic status. Irish vaccination data are used to determine the percentage of each age group that have received vaccinations for the infectious disease being modelled. For example, if 90%of 1 year olds in Ireland had been given the MMR vaccination in 2011 and we are running a model for 2012, we give each agent in the model with an age of 2 a 90%chance of having been vaccinated.  If an agent is vaccinated they are given a 97%chance of being immune to the disease. This takes into account vaccination failure and is based on the vaccine effectiveness rate for measles [13].  Half of the agents with age less than 1 are given immunity to measles to mimic passive immunity infants receive from their mothers [14]. For any agents that have an age corresponding to a vaccination year not in our data we give a 99%chance of being immune. Prior to vaccination campaigns the majority of the population would have either had or been exposed to childhood diseases, such as measles, leaving them immune in later life. This vaccination method is tailored to measles, to simulate a new or emerging disease we would have to use different assumptions about existing immunity levels”

123: are

Changed “is” to “are”

138: Do these also come from the CSO? These data are rare and will influence the model outcome, so the reader will naturally be curious about their origin.

Added the follow text to explain that the social networks in the model come from the synthetic population staring in lines 159-160:

“Social networks are included in the model. These are created based off of the simulated society and the schools and workplaces that agents are assigned to in the model.”

The full social paragraph starting on line 159 is now:

“Social networks are included in the model. These are created based off of the simulated society and the schools and workplaces that agents are assigned to in the model. Agents have a family social network that is made up of any agents living in their household. Agents also have a work or school social network that is made up of other agents in their workplace or their school and students are given an additional social network which is a class network that is made up of agents who are in their school and of the same age. Social networks help to determine the contacts an agent has in the model.”

142: probably replace "society" on 110 with synthetic population, as here.

We have left “society” in as it refers to a component of agent-based models for infectious diseases outlined in Hunter et al. 2017 and is used to describe the original town model in the Hunter et al. 2018 paper that is used as a base for this model.

148: agents' or agent, no ,

Changed agents to agents’

151: influence our results, holding

Added the comma after results

152: by the end of this paragraph, I'm not sure which one you're doing. You clarify in the next, but it might be worth just saying that here, or starting the next paragraph with hence or something. I'm left with a question and then have to figure out that the next paragraph answers it.

We have added extra text in the paragraph to show this, starting on line 165 with the new text in lines 178-180.

“In the Hunter et al. model, the population and thus the initial conditions vary slightly from run to run. For each run the model recreates the population again using the same probability distributions. This method allows for variation in the synthetic population and does not settle on a specific version of the population when the exact actual population is unknown. Although it might capture variability in the runs due to one particular set-up being more susceptible than others, there are some disadvantages of running the model this way. The first is time: model set-up can often take a large part of the runtime of the model. Holding a population constant can allow for a speedier set-up as the model does not have to make recreate the model environment on each run.  In addition, it increases the variability in the output making it difficult to attribute the difference in the output from the runs to agents' actions, or societal interventions, versus the variability of the disease itself. An alternative to this is creating the population once and then using the exact same initial population for each run.  If we are attempting to show how different interventions such as vaccination rates influence our results, holding the population steady allows us to more accurately attribute changes in our results to the interventions considered. Thus a constant population between simulation runs would be preferable. However, the effects of holding the population constant on the model results should be investigated in order to determine what impact holding the population steady will have on the model output.”

162: Generally, I would expect Tables 1 and 2 and 3 to belong to a results section. If not, if these are used to justify choices later, then there need to be some sentences in place saying something like: we see no difference between the static and resimulated populations, so in the final analysis we'll ______. In the middle of reading this, I cannot tell what's a final result and what's a step along the path

Tables 1, 2 and 3 are simulations that are run to help validate the scaled-up model. This validation is an important step in creating an agent-based model as it helps readers trust that the results from any experiments run using the model are valid.  We have added text in the beginning of the methods section to explain this starting on line 92 with the new text in line 96 and again in lines 97-100. 

“The Hunter et al.  [3] model uses openly-available data to model towns in Ireland.  The model is an agent-based model that was designed to simulate the spread of measles through an Irish town, however, the towns are considered in isolation with no commuting between towns. The model was validated in a number of ways: comparing the basic disease dynamics to expected disease dynamics, doing a sensitivity analysis on a number of key parameters and comparing the results to real data from measles outbreaks in Ireland. In this work, we focus on scaling up this model, from modelling towns in isolation, to take into account the interactions among populations in different towns within a region. In the following methodology sections we break our model down into the four main components of an agent-based model outlined in Hunter et al. [8]  and discuss the data used to create each component along with the assumptions necessary to scale up the model and the validation of these assumptions. These assumptions typically involve reducing the fidelity of different components of the model.  We then discuss an experiment that is done using the scaled up region model to look at how the centrality of a town influences the spread of an infectious disease through a network. The results of the experiment will be presented in Section 3.”

Additionally, we restructure the methodology section so that the sections dealing with scaling up the model are all subsections of Section 2.1 “Scaling up the Town model to a Region model” Section 2.1 includes the following new introductory text starting on line 102: 

“We take the following methodological approach in scaling up the model. For each of the four main components of an agent-based for infectious diseases (environment, society, transportation and disease) we consider assumptions that should be made in scaling up the model most of which involve reducing the fidelity of the model. Each assumption needs to be appropriately validated so that we know the changes and assumptions made did not change the underlying dynamics and function of the model. As the Hunter et al. model  has already been validated we aim to show through running simulations that the results of the scaled-up reduced fidelity model are not drastically different from the model we have already validated. The following sections discuss the assumptions made to scale each of the four main components of an agent-based model for infectious diseases and the validation of those assumptions.  Once all of the assumptions are made we additionally validate the entire regional model comparing the results of the town model to that of the regional model focusing on two specific towns.”

175-187: I'm confused by this. It seems that there are still subdivisions within the small areas (e.g. home vs work) and so they are not in the same physical location. It seems you've reduced the number of states within the small areas, but not completely - there is still the possibility of movement from discrete state to discrete state. Sure, you have reduced the dimension of the coordinates in the small area, but not so completely that they're all in the same location.

For the purpose of the simulation all agents in the same small area are physically coded on to the same environmental patch.  The subdivisions (home, work etc.) are not a physical change in location but a change in the agent’s variables.  If an agent moves from home to work in the same small area, they will not move in the simulation but will note that they are in a different location by changing their location variable. We have adjusted the text to reflect this. The paragraph is below starting on line 203. The new text is on lines 206- 208 and 210-211.

“As we described in Section 2.1.1, the agent-based model described in this paper differs from Hunter et al. by abstracting away from the small areas level of detail, and so agents do not move within a small area (although, importantly for this work, small areas may now be located in different towns) and the only transportation that occurs is between small areas. Each small area is represented by a single environmental unit or patch in the model.  Within a small area, all agents in the small area are physically coded in the same patch in the model but the agents keep track of their more abstract location within the small area. For example, an agent will know if they are at home, at work, at school or in the community and can differentiate between being in these different locations even though they will remain on the same patch.  Agents move between small areas but do not move around within a small area. However, all agents in the same small area are not in contact with each other at all times. Instead a variety of factors determine if two agents come into contact with each other. First, is the agent’s location, an agent at home will not come into contact with another agent who is at work. Second is an agent’s social networks. An agent will have a higher chance of coming into contact with a member of their family network in the community then a member of their class, school or work network with whom, in turn, they have a greater chance of coming into contact with than an agent who is not in any of their networks.”

220: Once recovered,              

Added the comma

224: Once infectious,

Added the comma

236: Scaled-up Model

Added the dash

256: Feels like results again.

The process of testing and validating the scaled up model is part of the methodology section as model validation needs to be done before experiments can be run with the model. The added to the first paragraph of the section to emphasize this, beginning on line 268 with the new text in lines 272, and 276-285:

“The extensions and variations of the Hunter et al  that we described above have been designed to enable us to scale the model up to simulate a geographical region, at a county scale, that contains multiple towns. As an initial validation of this scaled up model we decided to compare the results for simulations of outbreaks within two towns in the scaled up version of the model with the results for simulations of outbreaks in the same two towns when the towns were isolated within the simulation. Our expectation was that if the scaled up simulation was working appropriately then the outcomes of the simulations under these different conditions should be somewhat different but not drastically so.  If our simulations show drastic differences there may be a problem with how we scaled the model and the assumptions we made would have to be re-checked.  This initial validation test is an important step in the methodology of creating an agent-based model. Determining the best method to validate an agent-based model can be difficult because there is no set validation methodology [18]. One of the methods that is used to validate agent-based models is comparing the results of the model to a simpler validated model [8]. Skvortsov et al. [19] validate their agent-based model for infectious diseases in Australia by comparing the disease dynamics of their model to the disease dynamics produced by a SEIR equation based model. We propose doing something similar and comparing the results of our scaled-up region model to the validated town model.

294: or region, we

Added the comma

358: This is my major difficulty reading this paper: "for each set of runs." Which runs? I've seen a variety of simulations above, some with more influential choices than others, and the casual reader at this point will not be sure precisely which model we're using to make inferences.

The runs referenced have to do with the multiple times the model is run for each experiment to account for stochasticity.  To make this clearer we have changed the wording in a number of places. First in section 2.2 (formerly 2.6) where we discuss the experiments we change “types of runs” to experiments the adjusted paragraph starting on line 327:

“We use the scaled-up regional model to do two different experiments to look at how the centrality of a town in a network influences the spread of infectious disease in that town: in the first experiment, we the outbreak starts in a randomly selected small area within the county and then we look at where the outbreak spreads and how many outbreaks occur in each individual town; in the second experiment the outbreak starts in a given town and we again look at where the outbreak spreads. The two experiments are done to determine if the outbreak starting in a given town has an effect on the spread of the infectious disease. Each experiment is run 300 times to account for the stochasticity in the model.”

Additionally, we changed the first paragraph of the results section to reflect the change from runs to experiments and also to reference the experiments described in the methodology on lines 404-406. 

“The following sections describe the results found from the experiments on the influence of town centrality in a network on the spread of infectious diseases described in the Methodology Section 2.2. For each set of experiment we calculate the percent of runs \footnote{For each set of initial conditions in the experiments the models are run 300 times for stochasticity. The initial conditions that change between runs in the experiments is the starting location of the outbreak}  that lead to an outbreak (two or more cases of measles) occurring in the town. Table 8 shows the percent of runs that lead to an outbreak in each of the 16 towns when the outbreak starts at a random location in the county, or when it starts in one of eight different towns: Cloone, Dromahair, Fenagh, Kinlough, Leitrim, Manorhamilton, Mohill or Tullaghan.  Each column represents a different starting location of the outbreak.  The eight towns are selected to get a range of centralities but were also chosen using the town similarities data discussed in the previous section so that any differences found can be more easily attributed to the differences in centrality.”

The structure is clear-ish but leaves a lot out. I need to know: which choices do we have to make when getting to the final model, and which choices have we actually made, and why? It's fine if you put the various tables above in the methods section to justify the various simplifications you've made, but the first paragraph of the results section needs to recall all of this and list what was done where. I need to know precisely what the final model is that is being run and analyzed.

It may be useful to have a table describing the orthodox Hunter [3] model and the version used here. The reader will lose track of what has been changed, after all the effort you put into demonstrating that the changes were feasible. Let me know what you're changing, why, why it's alright (your simulations) and then recap.

In Section 2.1.5 (formerly Section 2.5)  starting at line 268 with new text starting at 270 we add the following text and table

“The extensions and variations of the Hunter et al. [3] model that we described above have been designed to enable us to scale the model up to simulate a geographical region, at a county scale, that contains multiple towns. The key differences between the hunter et al. [3] model and the scaled up model can be found in Table 4.”

Additionally at the end of Section 2.1.5 we added the following text in the paragraph starting at line 302 with the new text starting on line 312:

“From the results we can see that the town only model that allows for commuting results in fewer outbreaks than the town only model where agents cannot leave the town. This makes sense as if the infected agents are commuting outside of the town, once they are outside the town they do not come into contact with other agents and thus cannot spread the disease until they return to the town. In addition, the county model results for both towns are somewhere between the completely closed town model and the town model with commuting.  Again, this makes sense as in the county model the agents are not restricted to staying within their town so there is a smaller chance of an outbreak in the town in the county model because in some cases the infected agent will commute out of the town and take the infection with them. The outbreak percentage is, however, higher than for the town model that allows commuting because there are other agents in the model who can become infected keeping the outbreak going. We take this as a sign that the county model is working as it should be. Based on our analysis and validation work in this and previous sections, in the scaled-up regional model we will use the model characteristics outlined in Table 4: the environment is created using a single patch for each small area, the society is kept constant each run and transportation is based off of a gravity model.” 

404 and elsewhere: in-degree

Changed “in degree” to “in-degree” where it occurs

418: same

Changed “in degree” to “in-degree” where it occurs

438: Conclusions such as this are probably best moved to a discussion.

We moved  the text:

“Compared to the correlations with in-degree centrality the correlations for out degree centrality are markedly different than those with total degree centrality.  One possible conclusion from this is that agents commuting into a town are more important than agents commuting out of a town for the spread of an infectious disease through a network.”

To the second paragraph of the discussion section starting on line 590 with the moved text in lines 593-598:

“Modelling how agents movements influence the course of an outbreak is important in studying how to react when an outbreak occurs. If an outbreak starts in a given region, towns that are more susceptible in that region can be a focus of the response, with more resources sent to these towns. Our results showed that compared to the correlations with in-degree centrality the correlations for out degree centrality are markedly different than those with total degree centrality.  One possible conclusion for this is that for the spread of an infectious disease through a network agents commuting into a town are more important than agents commuting out of a town. Therefore, the planning of an intervention to stop the spread of a disease to a town should focus on who commutes into a town versus those who commute out. In addition, the location of the initial cases can be used to guide responses. For example, if the outbreak starts in a town with high in-degree centrality closing the town might prevent further spread of an outbreak while if the town has low degree centrality the best course of action might be closing things such as schools or workplaces in nearby towns with high degree centrality as this would stop the outbreak from spreading into the high degree centrality town and then to many other locations from there. Additional simulations could be run including such restrictions to better understand how they could help to reduce the size and severity of an outbreak.”

Round 2

Reviewer 1 Report

Author reply’s letter is enough to accept de article 

Author Response

We would like to again thank the reviewer for their thoughtful and insightful comments on our paper.  The comments allowed us to improve our work and we have integrated their suggestions in to our revised version of the paper.

Sincerely,

Elizabeth Hunter

Reviewer 2 Report

I'm startled - seeing the quick response, I assumed most of the responses would be "nah." However, my difficulties understanding the flow of the paper, especially with which model is being considered where, were all dealt with appropriately. Table 4 is precisely what I needed and ties everything together.

As far as I can tell, all that remains is just small bits, e.g.

in-degree and out-degree

Thus,

Especially early on in the pandemic, agents' actions

additional data were used

Author Response

We would like to again thank the reviewer for their thoughtful and insightful comments on our paper.  The comments allowed us to improve our work and we have integrated their suggestions in to our revised version of the paper. Please find below our point by point response discussing how we addressed each of the referees new comments.

Sincerely,

Elizabeth Hunter

As far as I can tell, all that remains is just small bits, e.g.

in-degree and out-degree

We changed all cases of “out degree” to “out-degree”.

Thus,

We added commas after “Thus” in two instances.

Especially early on in the pandemic, agents' actions

We added the comma between pandemic and agents’.

additional data were used

We changed “additional data, however, was used” to “additional data were used”